# A New Correction Theory and Verification on the Reducing Rate Distribution for Seamless Tube Stretch-Reducing Process

**Jianhua Hu \*, Sheng Yang, Yulong Huang, Xiaohua Wang and Jianxun Chen**

School of Materials Science and Engineering, Taiyuan University of Science and Technology,
Taiyuan 030024, China
* Correspondence: 2005022@tyust.edu.cn; Tel.: +86-139-342-031-33

**Abstract:** The reducing rate distribution is critical for the quality and precision of the final pipe during the process of stretch-reducing of the seamless pipe. The inhomogeneous deformation of the pipe may occur if the reducing rate distribution is improper. This paper analyzed the trend of the reducing rate distribution in terms of metal flow and put forward a "three-point and two-section converged" correction theory based on relevant research. In order to verify the theory, the finite element model is established according to the results obtained from the modified model. The simulation is accomplished successfully, and the cross-section of the pipe is evenly reduced with the longitudinal metal flowing uniformly. The result from the experiment is consistent with that from the simulation, which shows the rationality of this theory, providing a new method for the reduction rate allocation.

**Keywords:** stretch-reducing; seamless pipe; three-point and two-section converged; reducing rate distribution; finite element model; correction theory





## 1. Introduction

According to Li and Gao (2016), for the stretch-reducing mill line, the maximum range of the total reducing rate was 75–80%, wall thinning rate 35–40%, and elongation coefficient up to 6–8 [1]. Based on the change in tension between stands, the stands may be divided according to Wang and Qi (2013) into three parts: rough rolling, intermediate rolling, and finishing rolling [2]. Generally, the rough rolling uses 2–3 tension-rising stands. Sun and Zhang (2012) pointed out that with the increase in stretch-reducing mill stands, the reducing rate rising stands were increasing accordingly, and thus, the tension was increasing gradually [3]. The intermediate rolling uses working stands, where the outer diameter and wall thickness may be reduced greatly. In other words, most of the work of the stretch-reducing mill is completed by this part, and generally, the reducing rate was unchanged or gradually decreased. Huo et al. (2016) and Li et al. (2015) agreed that the finishing rolling uses tension dropping stands, also known as stands for the finished product, for which the rolling reduction was generally not large [4,5].

The purpose of stretch reducing is to make the outer diameter and wall thickness meet relevant requirements for a finished product. In order to avoid transverse wall thickness inhomogeneity and excessive load of tubes, the rolling reduction for each stand must be within a reasonable range [6]. For a tube with a small outer diameter and wall thickness, a large reducing rate may be selected, but the quality must be guaranteed, while for a tube with a large outer diameter and wall thickness, a small reducing rate should be selected to avoid inner polygons and excessive load.

In the production of a small-diameter steel tube, a large reducing rate should be adopted, but an excessively large reducing rate may cause serious deformation, resulting in an uneven wall thickness of the steel tube. Additionally, excessive reducing may affect the stability for steel tube rolling, cause ear folding, etc., and even rolled stock jamming. If wide ranges of tube diameters and wall thicknesses are produced by a mill line, before

reducing the rate distribution of stands, there must be several series of passes divided according to the outer diameter and wall thickness of the tube based on the product mix. It may be divided into two series according to the size of the product. Series A is used for producing small outer diameters and small-to-medium wall thicknesses of tubes, while Series B is used for producing large outer diameters and large wall thicknesses of tubes. Therefore, different specifications of tubes should match each other within passes of the corresponding series dependent on the outer diameter and wall thickness [7].

The reducing rate distribution may influence the subsequent design of passes and distribution of tension greatly, therefore, a reasonable distribution is very important. So far, most of the reducing rate distribution theories for stretch-reducing mill lines result from the experience of predecessors obtained in the process of localization after introducing relevant devices from other countries. Sun (1999) used the power function to establish the general equation of the reduction rate distribution of tension reduction machine working units, but the real effect was not verified [8]. Ding et al. (2013) had designed the pass the of Baoshan Iron and Steel Co. 28-stand tension reduction machine with a power function reduction rate. The reduction rate of each stand was designed in the form of a power function with the new pass, which increases the load of the motors and reduction boxes in groups 1 and 2 of the tension-reducing machine, especially the damage to the motor in group 2 [9]. Therefore, the relationship between the design method of the pass and the actual production needs to be balanced comprehensively when designing the pass of the tension-reducing machine with this method. Shen et al. (2017) employed the power function distribution general formula for the reduction rate of the working stands of a tension-reducing mill, so that the logarithmic reduction rate of each stand of the working unit decreases with the increasing number of the stands [10]. After the pass rolling of the tension-reducing mill designed with this method, the outer diameter of the steel tube meets the requirements but the error is not very satisfying.

This study analyzed the law of reducing the rate distribution in terms of metal flow and put forward a "three-point and two-section converged" correction method based on relevant experience. A new reasonable methodology for the reducing rate calculation was established based on corresponding conditions and deformation scenarios. Compared with the calculated results based on a previous empirical formula, the results of the correction method showed smooth curves, which are conducive to metal flow.

## 2. Conventional Reducing Rate Distribution Theory

At present, a three-high stretch-reducing mill is adopted in most of the mills, with different reduction at each stand. The schematic diagram is shown in Figure 1.

There are two main methods for reducing rate distribution: the decrement method and constant method. For the absolute reduction of a single stand, the decrement method is adopted. See Figure 2.

Relevant literature (Zhang et al., 2006 and 2008) has shown that, at present, the reducing rate distribution calculation is based on two reducing rate distribution methods of the Baosteel 114 mill line. Method 1: reducing rates of Stands 1, 2, and 3 are fixed at 0.02, 0.03, and 0.04, and the empirical formula is adopted for falling stands [11,12]. However, for a different number of stands, it is not as reasonable to adopt the same fixed reducing rates for different reductions; Method 2: AB protocol mentioned above [13]. Figure 3 is the reducing rate curve obtained by calculating a mill line according to the conventional equal reducing rate distribution method presented in relevant literature. To further clarify the curve, it was partially used for local amplification. This was obtained using a five-stand reducing rate rising stand calculation based on the A series [14,15]. Obviously, the single-stand reducing rate for rising stands was too small, and thus, the single-stand reducing rate of working stands was too large. The big leap in the reducing rate between the rising stand and the working stand caused uneven deformation. Therefore, sometimes, this conventional distribution method is not suitable.

In the calculation of reducing rate distribution, there is no fixed value, and thus, it is difficult to select relevant parameters. If the calculation of results is incorrect for inappropriate initial parameters, it is necessary to constantly correct the initial conditions and modify the calculation process. Therefore, when the reduction rate of one of the stands is improper, the reduction rate of the subsequent stands would be influenced and the error would accumulate, resulting in uneven deformation so that the final pipe is not qualified [16–18]. Thus, a new modified model is proposed, which is called the "three-point and two-section converged" correction approach.

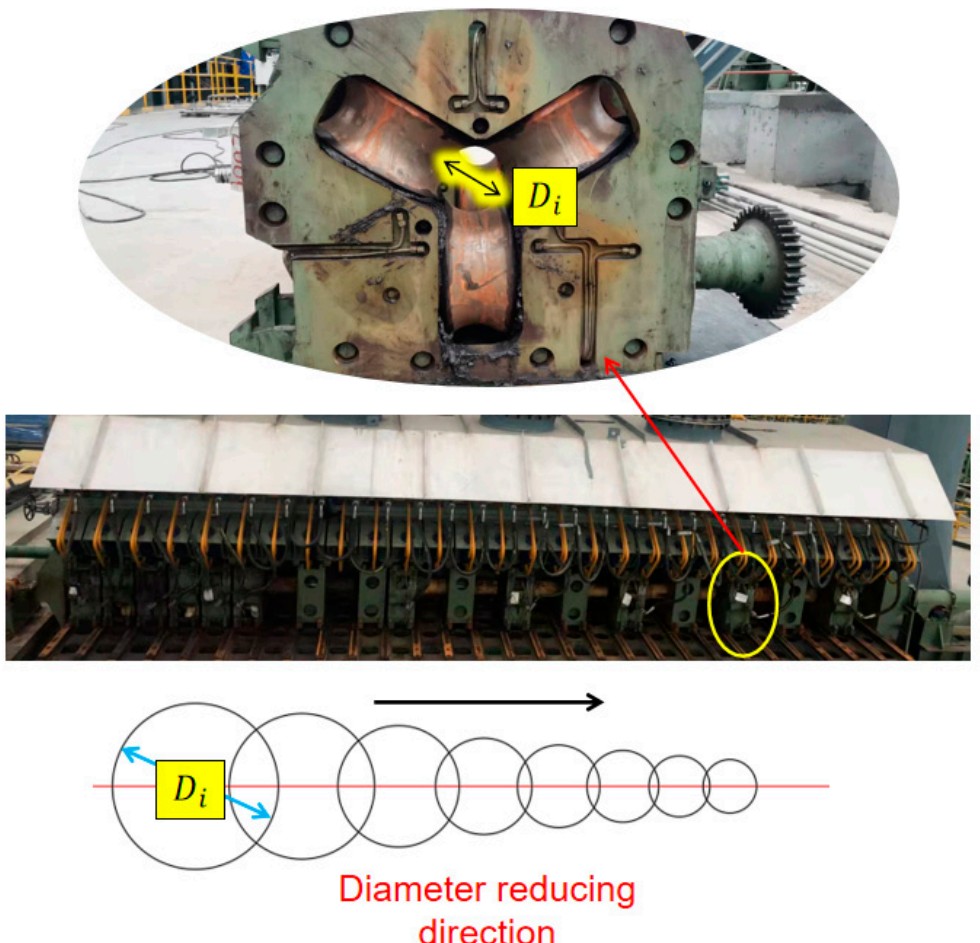

**Figure 1.** Device diagram of three-roll stretch reducing.

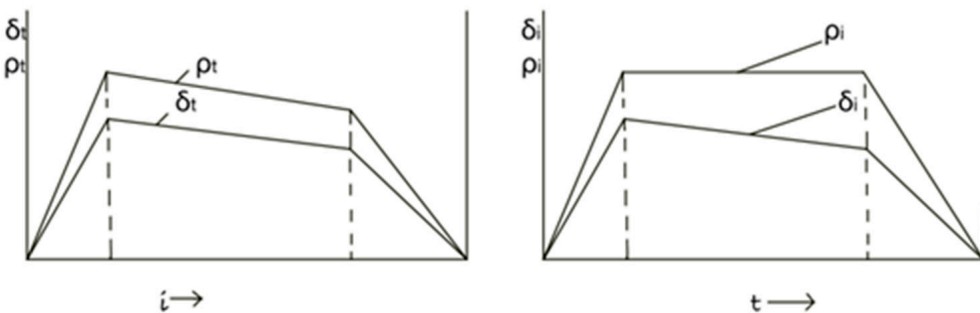

**Figure 2.** Conventional method for reducing rate and reduction distribution.

**Figure 3.** Reduction rate distribution method from conventional experience.

### 3. "Three-Point and Two-Section Converged" Correction Approach

*3.1. Basic Method*

Relevant observation and analysis revealed that there is a correlation between the reducing rates of stands, but the conventional empirical formula fails to reflect this correlation. Therefore, this study put forward the "three-point and two-section converged" approach. In the reducing rate distribution curve, after the most important three points (i.e., points for the first stand of the reducing rate, the maximum reducing rate, and the last working stand of the reducing rate) are determined, the curve equations of two sections (i.e., rising stand section and working stand section) are determined accordingly. Therefore, it is three points and two sections that determine the whole reduction rate method, which can be solved using two points and an initial condition. In addition, the three points are interrelated with all other points, namely they influence each other. All points are linked together through the two curve equations. In fact, the three points determine the positions of all other points indirectly, and, therefore, they are "converged".

This approach listed some basic theoretical conditions for reducing rate distribution:

(1) Gripping. The initial reducing rate of stand shall not be too large, otherwise the gripping will fail.

(2) Limitation on the maximum reducing rate. If the reducing rate is too large, the rolling may be unstable, thus, causing problems such as uneven wall thickness.

(3) Uniform decline in the reducing rate for a working stand. The reducing rate decreases progressively because with the rolling, the closer the distance to the end of the stand and the lower the temperature.

(4) Limitation on the reducing rate of the last working stand. Under the limitation on the maximum reducing rate, the reducing rate of the last working stand shall not be too small for sufficient reduction.

According to the stretch-reducing theory, in the process of production, with the increasing number of stands for rolling, the increase in deformation degree, the increase in rolling speed, and the decrease in rolling temperature will increase both the deformation resistance and friction coefficient, thus, increasing the rolling force and energy consumption and tool wear as well. From this perspective, the diminishing distribution of the reducing rate is

more reasonable [5]. Therefore, in this approach, the reducing rate increases uniformly in the rising section, and decreases uniformly at the working stands.

Based on the basic theory, a new correction method was proposed for establishing equations for the two curve sections. Assuming the reducing rate equation of a single stand for the rising stands:

$$y = a_1x + b_1 \tag{1}$$

and the reducing rate equation for the working stands:

$$y = a_2x + b_2 \tag{2}$$

where $a_1$, $a_2$, $b_1$, and $b_2$ are the undetermined coefficients.

As shown in Figure 4 (assuming five finishing stands), by substituting three-point coordinates and initial conditions, two curve equations may be obtained, and thus, the reducing rate value of each stand may be determined.

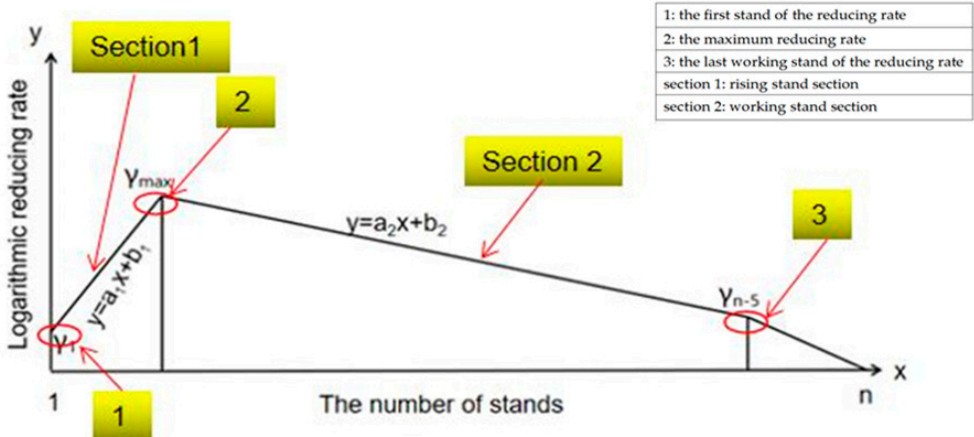

**Figure 4.** Basic three-point and two-section method.

In the rising stand section, based on experience, the logarithmic reducing rate of the first stand is generally 2–3%, therefore, it can be regarded as a known value.

In the working stand section, according to the reducing rate of the first finished product stand, the reducing rate of the last working stand can be determined approximately.

In addition, the total logarithmic reducing rate of the rising stands and the working stands can be obtained, therefore, the maximum reducing rate can be solved by combining the two equations. If the maximum reducing rate meets relevant requirements, the reducing rate of each stand can be determined.

### 3.2. Solution Based on the Three-Point and Two-Section Converged Approach

After summarizing the whole solution, the procedure for determining the reducing rate of each stand is obtained as follows:

(1). Determine the total logarithmic reducing rate according to the diameters of the pierced billet and finished product.

(2). Determine the minimum number of stands according to the maximum reducing rate ($\gamma_{max}$) of a single stand.

(3). With the average logarithmic reducing rate generally between 0.04 and 0.05, determine the number of stands according to the minimum number of stands and the range of the average logarithmic reducing rate.

(4). Determine the reducing series to be used according to the number of stands as well as the diameter and wall thickness range of the finished product.

(5). Determine the initial number of stands and the number of stands for the finished product.

(6).  Determine the total reducing rate of finished product stands according to the number of finished product stands.

(7).  Determine the reducing rate of each finished product stand according to the total reducing rate of the finished product stand, calculate the maximum slope of the line segments between reducing rate points, and thus, calculate the reducing rate of the last working stand according to the slope.

(8). Assume the reducing rate of the first stand according to the commonly used range.

(9).  Establish the two-section curve equation set for the tension rising stand and working stand.

(10). Solve the maximum reducing rate.

(11). If the solved maximum reducing rate is lower than the rated maximum reducing rate, calculate the reducing rate of each stand directly, or return to reset the reducing rate of the first stand, and reduce the maximum reducing rate.

The procedure is represented as a flow chart for the convenience of understanding as shown in Figure 5.

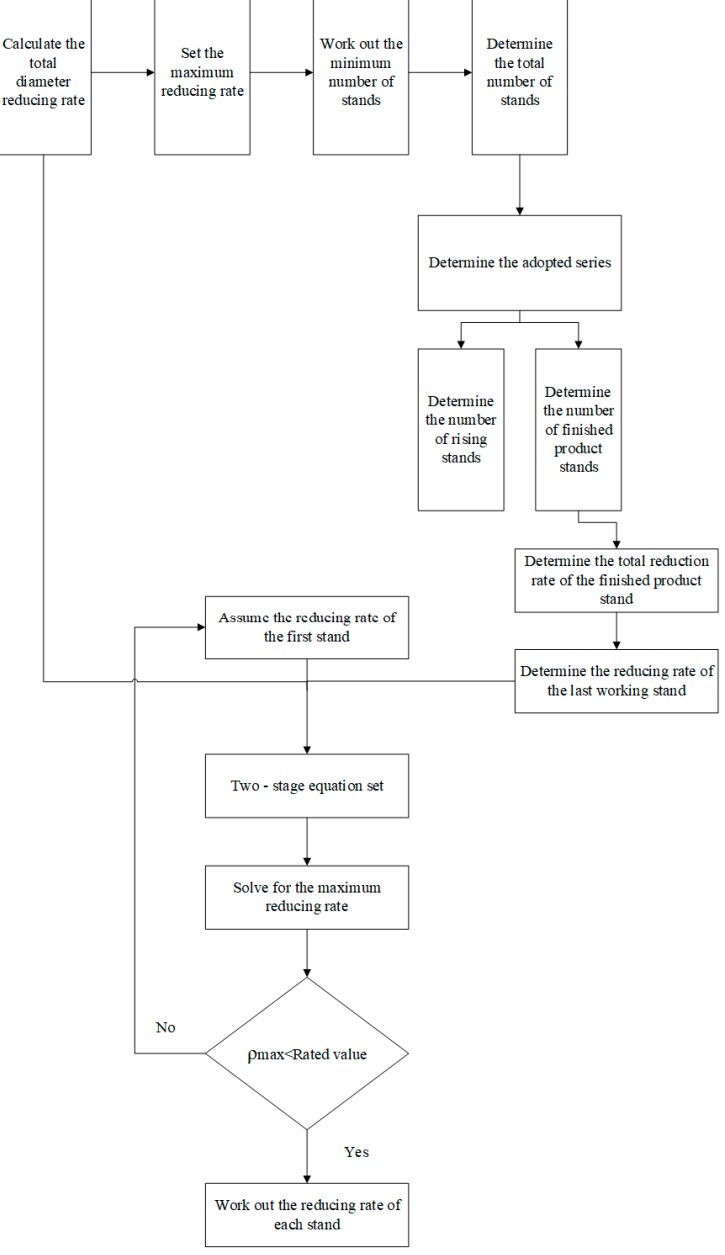

**Figure 5.** Flow chart for the solution.

*3.3. Example Calculation*

The specific procedure for calculating the "three-point two-section converged" correction method is presented below with the actual working condition of an enterprise as an example. The diameter of the mill line after shutting out from the extracting mill is 118 mm, the diameter of the finished tube is 38 mm, and the wall thickness is 5 mm. The total number of stands is 28.

Based on experience of the enterprise in the division of the pass series, there are two series according to different reducing rates:

A series—for the production of small-sized steel tubes, a series of large reductions, for which 5 rising stands and 5 dropping stands are selected, and the maximum relative reduction for each stand: $\rho_{max}$ = 8.00%.

B series—for the production of large-sized steel tubes, a series of small reductions, for which 3 rising stands and 4 dropping stands are selected, and the maximum relative reduction for each stand: this was a small sized case, therefore, 5 rising stands and 5 dropping stands were selected.

According to the diameters of the pierced billet and finished tube, the total logarithmic reducing rate $\gamma_{\sum}$ was determined as follows:

$$\gamma_{\sum} = \ln \frac{d_k}{d_r} = \ln \frac{118}{38} = 1.133 \tag{3}$$

Here,

$d_k$—the initial diameter of the pipe before the reducing process.

$d_r$—the diameter of the pipe after the reducing process.

According to Li (2006), for 5 finished product stands,

$$d_m/d_r = 1.06 \sim 1.10 \tag{4}$$

$d_m$—the diameter of the last working stand.

Accordingly, the total reducing rate of the finished product stands $\gamma_f$ was calculated as follows:

$$\gamma_f = \ln \frac{d_m}{d_r} = 0.0583 \sim 0.0953 \tag{5}$$

$$\gamma_z + \gamma_a = \gamma_{\sum} - \gamma_f = 1.1331 \, (0.0583 \sim 0.0953) = 1.0748 \sim 1.0378 \tag{6}$$

$\gamma_z$—the total reducing rate of the working stands.

$\gamma_a$—the total reducing rate of the rising stands.

Considering the reason for temperature reduction, the average reducing rate of the finished product stand should be smaller than that of the rising stand and working stand, and the smaller it is the more it is within a reasonable range, that is, for the total reducing rate of the rising stand and working stand, the upper limit (1.0748) should be adopted, and accordingly, $\gamma_f$ = 0.0583.

The following was obtained:

$\gamma_{28} = 0$.

$\gamma_{27} = 0.05 \times \gamma_f = 0.0003$.

$\gamma_{26} = 0.2 \times \gamma_f = 0.0117$.

$\gamma_{25} = 0.3 \times \gamma_f = 0.0175$.

$\gamma_{24} = 0.45 \times \gamma_f = 0.0262$.

$\gamma_i$—the reducing rate at the *i*th stand.

The whole dropping part was divided into 5 segments of straight lines, and with the maximum slope as the approximate slope, $\gamma_{23}$ was determined with an approximate value: 0.035.

For the first rising stand, the distribution rate $\rho_1$ was 2~3%, therefore, the following was obtained:

$$\gamma_1 = \ln \frac{1}{1 - \rho 1} = 0.0202 \sim 0.0304 \tag{7}$$

The lower limit was adopted for the convenience of gripping.

Relevant equations were established for the rising stand and working stand and points $(0, 0.02)$, $(5, \gamma_{max})$ were substituted into Equation (1). Thus, the following was obtained:

$$\gamma_{max} = \gamma_6 = 5a_1 + 0.02 \tag{8}$$

The coordinates of the start point and end point of the working stand section, $(5, \gamma_{max})$ and $(22, \gamma_{23})$, were substituted into Equation (2). Thus, $a_2$ and $b_2$ expressed as $a_1$ were obtained.

The reducing rate of each stand was given, and then, the reducing rates of working stands and rising stands were summed up to establish an equation for $a_1$.

The reducing rate of each stand, as well as the ideal diameter and reduction of each stand, was obtained as shown in Table 1:

**Table 1.** Distribution of the reducing rate and reduction of 28 stretch-reducing mill sets.

| NUMBER OF STANDS | 1 | 2 | 3 | 4 | 5 | 6 | 7 | 8 | 9 | 10 |
|---|---|---|---|---|---|---|---|---|---|---|
| LOGARITHMIC REDUCTION RATE | 0.02 | 0.0287 | 0.0375 | 0.0462 | 0.0549 | 0.0636 | 0.062 | 0.0603 | 0.0586 | 0.0569 |
| AVERAGE DIAMETER (MM) | 115.7 | 112.4 | 108.3 | 103.4 | 97.85 | 91.82 | 86.3 | 81.3 | 76.63 | 72.39 |
| REDUCTION (MM) | 2.337 | 3.275 | 4.132 | 4.886 | 5.523 | 6.033 | 5.515 | 5.05 | 4.623 | 4.238 |

| NUMBER OF STANDS | 11 | 12 | 13 | 14 | 15 | 16 | 17 | 18 | 19 | 20 |
|---|---|---|---|---|---|---|---|---|---|---|
| LOGARITHMIC REDUCTION RATE | 0.0552 | 0.053 | 0.0518 | 0.0502 | 0.0485 | 0.0468 | 0.0451 | 0.0434 | 0.0417 | 0.0401 |
| AVERAGE DIAMETER (MM) | 68.5 | 64.93 | 61.651 | 58.635 | 55.860 | 53.306 | 50.955 | 48.790 | 46.796 | 44.958 |
| REDUCTION (MM) | 3.888 | 3.570 | 3.2806 | 3.0162 | 2.7746 | 2.5536 | 2.351 | 2.1653 | 1.9945 | 1.837 |

| NUMBER OF STANDS | 21 | 22 | 23 | 24 | 25 | 26 | 27 | 28 | | |
|---|---|---|---|---|---|---|---|---|---|---|
| LOGARITHMIC REDUCTION RATE | 0.0384 | 0.0367 | 0.035 | 0.0262 | 0.0175 | 0.0117 | 0.0030 | 0 | | |
| AVERAGE DIAMETER (MM) | 43.266 | 41.71 | 40.27 | 39.231 | 38.5511 | 38.103 | 37.989 | 37.989 | | |
| REDUCTION (MM) | 1.6923 | 1.558 | 1.434 | 1.041 | 0.6806 | 0.4484 | 0.1141 | 0 | | |

The maximum reducing rate was verified as less than 8%, which is within the range of the maximum reducing rate. Thus, the calculation for the method is completed.

## 4. Verification of the Finite Element Simulation and Experiment

According to the modified theory, the finite element model is established, on the basis of the average diameter of the tube obtained with 28-stand stretch reduction. The initial speed of the pipe at the entrance of the reduction unit, $V_0$, is set as 1.674 m/s, and a pusher is used to help the initial bite of the rolled piece. According to the diameter of the pipe in each pass, the cross-sectional area of the pipe is calculated, then the exit speed of the pipe can be obtained so that the roll speed of each stand can be obtained.

In the pre-processing conditions, the original length of a section of pipe is 1200 mm, the outer diameter is 129 mm, and the wall thickness is 13 mm. The number of nodes is 12,199 and the number of elements is 36,782. The material selected was 45 steel and the temperature was set as $1000°$. The flow stress curve model, $\overline{\sigma} = \overline{\sigma}\left(\overline{\varepsilon}, \dot{\overline{\varepsilon}}, T\right)$, was selected from the simulated software, and Von Mises was used as the yield criterion for the simulation. The contact friction coefficient was taken as 1.35 between the rolls and the tube, and the thermal conductivity was selected as 11 (N/s/mm/C). As shown in Figure 6, the simulation was performed.

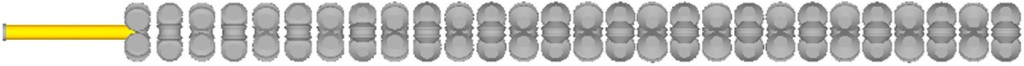

**Figure 6.** The finite element model of the 28-stand tension-reducing mill.

By analyzing the finite element simulation results obtained by this modified model, it can be seen that after the pipe enters each stand, the metal flow is stable, the deformation

is almost composed of the longitudinal elongation, and the transverse movement of the metal can be ignored. As shown in Figure 7, the equivalent strain diagram of step 180 of the finite element simulation is intercepted. With the comparison of the right numerical bar, it can be concluded that the equivalent strain increases gradually with the tension reduction process. The maximum value of equivalent strain is 0.985 mm/mm and the minimum value is 0.0198 mm/mm. The pipe is separated along the longitudinal direction at the center of the pipe. From the longitudinal section, it can be shown that after being rolled by each stand roll, the average diameter of the rolled piece uniformly decreases with the increasing stand, which means the distribution of reduction rate and reduction amount is reasonable, i.e., conforming to the reduction law.

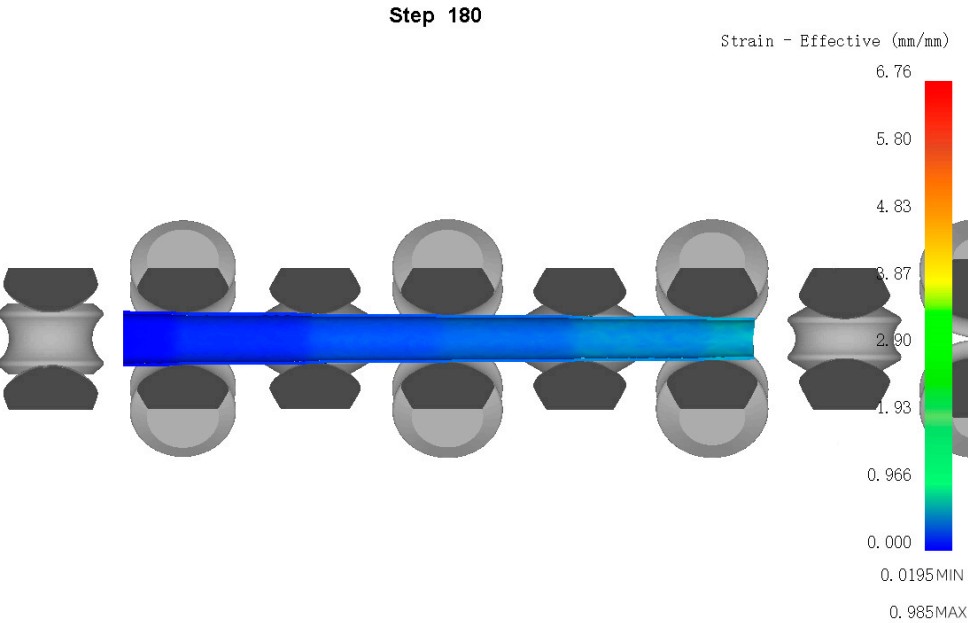

**Figure 7.** Equivalent strain distribution of the finite element model (step 180).

The part of the seamless pipe which was reduced between 15–22 frames at the step of 365 is shown in Figure 8 below. Figure 8a on the left side is the equivalent stress sectional view of the pipe fittings at step 365 of the finite element simulation. The yellow band at the outer of the pipe outline represents the wall thickness, which shows that the wall thickness changes basically evenly during the whole rolling process. When the tail section of pipe fittings bite into the 15th stand, it can be seen that the equivalent stress is concentrated and sharply increased during rolling at each roll, forming a stress field, which agrees with the plastic flow law of metal. The Figure 8b on the right refers to the equivalent strain of the pipe fittings at step 365 of the finite element simulation. By comparing the color bars, it can be seen that the equivalent strain is steadily increasing without any sudden leap, and the corresponding pipe walls are becoming thinner and thinner, extending longitudinally continuously, conforming to the law of metal plastic flow.

However, in the process of production using the traditional model, there is a jump in the reduction data at the 19th rack of the tension-reducing mill. An inspection of the rolled object in traditional model production shows that there is a larger reduction in the diameter on the left side of the tube compared to the right side, as shown with the red mark in Figure 9, producing a step-like mark there, i.e., there is an obvious leap in the diameter which is not desirable in the real production. After using the modified model, it is found that the decreasing degree of the reducing rate is lower and there is no obvious step trace at the 19th, the 20th, and subsequent stands. Compared with the traditional model, the step trace is optimized, which shows that the adjustment of the process parameters is more conducive to the uniform flow of metal.

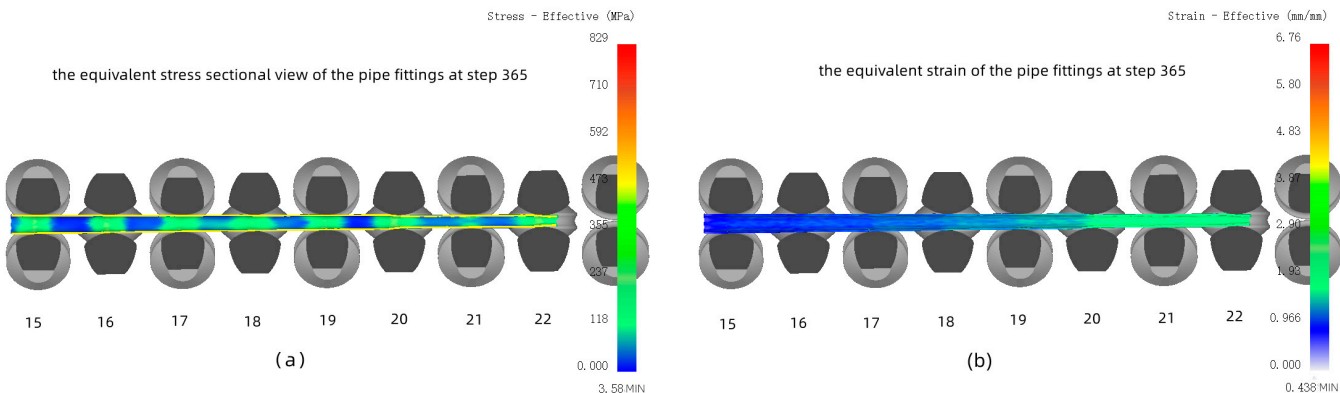

**Figure 8.** The equal effect diagram and equivalent strain diagram of part of the 28 tension reducing machines.

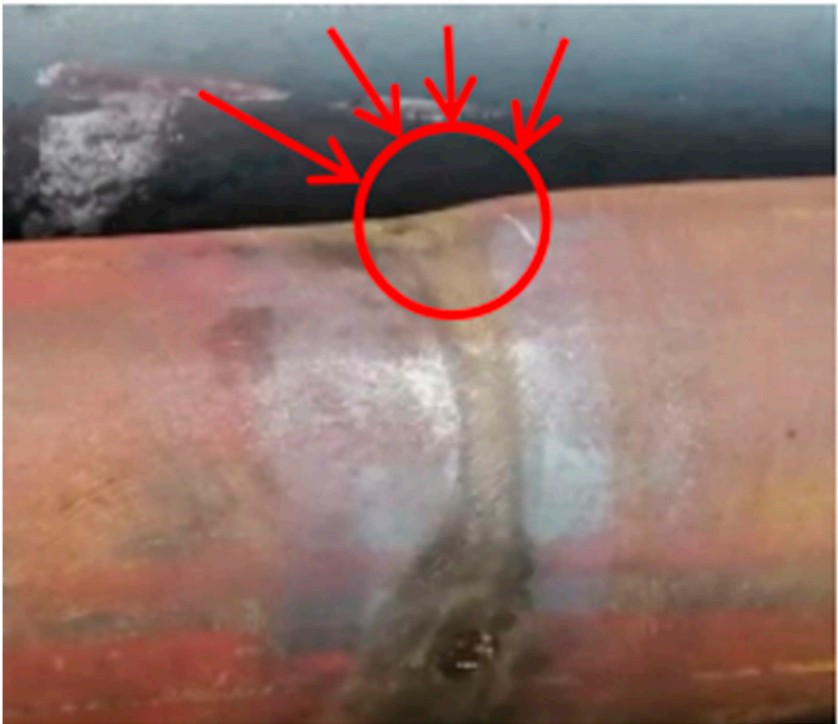

**Figure 9.** Interception of the traditional model of the rolling stock physical object.

Tension reduction easily causes tail triangles and other defects, which can be explained by metal flow. In order to avoid this phenomenon, the parameters from the modified model are employed and a cross-section of the pipe during the simulation were used for analysis. As shown in Figure 10, the partial cross-sectional views, Figure 10a–d, are taken when the tube blank is dragged into the second stand, the fourth stand, the sixth stand, and the eighth stand during the finite element simulation of tension reduction (because the number of stands is too large to show them all, several cross-sectional views with obvious characteristics were selected here for explanation). Figure 10e,f display the cross-section outline after the pipe enters the 20th and 21st racks during the finite element simulation of tension reduction. From Figure 10, it can be clearly seen that after the model is optimized by the modified method the cross-section is almost circle and the plastic flow of metal is obviously improved. The wall thickness of each cross-section is uniform and there is no obvious tail triangle effect.

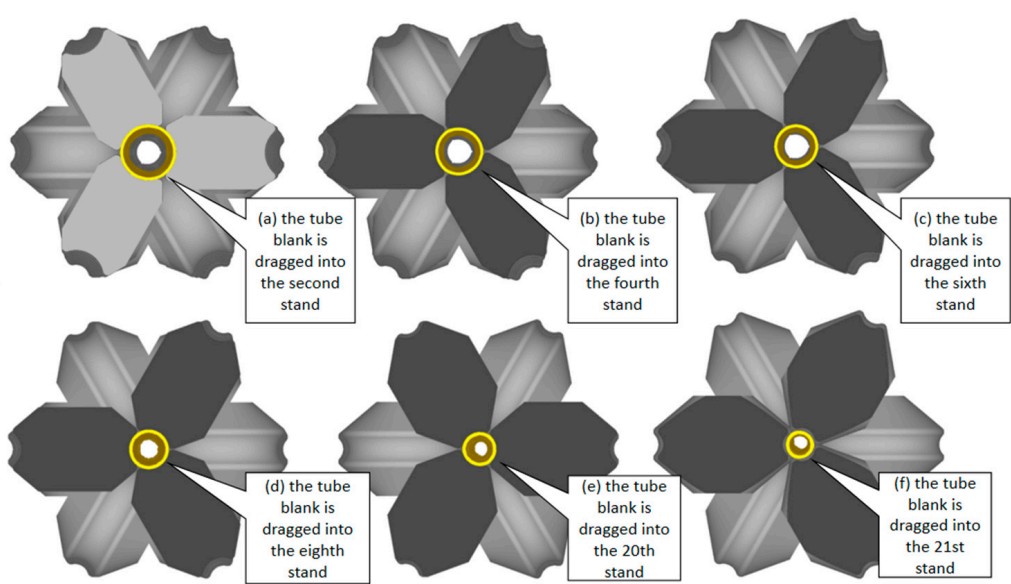

**Figure 10.** Partial cross-sectional view.

Using the above finite element simulation, the feasibility of this modified algorithm can be proved. Using this model for production, the pipe obtained during the process of rolling is revealed in Figure 11 below, which is close to the finite element simulation result. Obviously, the distribution of the reduction rate and the reduction amount are reasonable. In the last few stands, the diameter reduction is very small, which basically trends towards the final size.

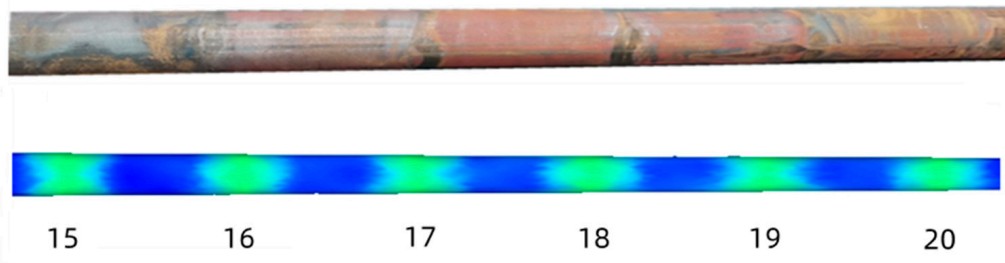

**Figure 11.** Modified model simulation and object plug comparison.

## 5. Conclusions

(1) Considering the irrationality of the traditional empirical formula, the theory of the "three-point and two-stage integration" modified model was put forward, and the calculation ideas and concrete calculation steps were provided, and it was explained in detail using a concrete example. The data are real, full, and accurate, which is beneficial for further research.

(2) Using the results obtained with the "three-point and two-stage integration" modified method, the finite element model was established. From the simulation results, the longitudinal flow of metal was uniform, so the modified theory was verified. In addition, in a real tension-reducing mill, the calculation results from the modified model were adopted to produce the pipe as well. Comparing the pipes from the finite element simulation and real production, the wall thickness, average diameter, and reduction situation were basically consistent, indicating the rationality and feasibility of the modified theory.

**Author Contributions:** All authors contributed to the study. Method design and analysis were performed by J.H., S.Y. and Y.H.; The first draft of the manuscript was written by Y.H.; the review and revision of the first draft was completed by J.C.; X.W. collected the experimental data employed in the manuscript. All authors have read and agreed to the published version of the manuscript.

**Funding:** This paper is funded by the Major Project of Ministry of Science and Technology of Shanxi Province, China (No.20191102009) and Science and Technology Key Development Project of Shanxi Province, China (201903D121049).

**Informed Consent Statement:** Informed consent was obtained from all subjects involved in the study.

**Data Availability Statement:** The datasets used during the current study are available from the corresponding author on Jianhua Hu.

**Conflicts of Interest:** The authors have no relevant financial or non-financial interests to disclose.

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
