# Peer review of "A New Correction Theory and Verification on the Reducing Rate Distribution for Seamless Tube Stretch-Reducing Process"

_crystals, doi:10.3390/cryst12091296_

Round 1

Reviewer 1 Report

 A new correction theory and verification on the reducing rate distribution for seamless tube stretch-reducing process

Abstract:

1. Line 12, after pull-stop, there must be ‘single space’- Authors must modify

2. In line 15, there were two dots, one has to remove

Introduction

3. Literature in the document must be written in past tense, but most of the literature was written in the present tense, should be re-written entire part. Also, literature explanation is inconsistence, not concluded them and showed the literature gap.

4.  2 conventional reducing rate distribution theory…..first letter ‘c’ must be capital

5. Fig. 1 is not a schematic diagram, modify it and show the reference from where the figure is taken; (Authors should know the difference between actual diagram and schematic diagram).

6. Reference citation is inconsistence, should be maintained uniformly everywhere in the document.

7. Authors not put ‘.’, after the Figure captions. Also, Either ‘Figure’ or ‘Fig.’ use in the document.

8. “Therefore, the formula is local, without global 111 consideration. A false step makes all steps wrong, and so, relevant design is both difficult 112 and complex.” How authors decided local and global, discuss more clearly.

9. 3". Three-point and two-section converged" correction approach….should be 3. "Three-point and two-section converged" correction approach…modifies it

10. As per the mathematical calculations, reduction ratio is not uniform, when it comes to FEM analysis material flow is claimed as uniform……..clarify it

12. I general, FE analysis solution will be decided based on the pre-processing conditions like yield criterion, mesh size, flow curve conditions…none of them were discussed here. Authors can incorporate these issues.

13. In Fig. 9, what authors want to explain with red marks indicated on it?

14. Conclusions looks very general, authors must modify with scientific outputs with logically.

The overall content of the paper is looking good, but needs more corrections and consistency with the explanations of considered content and obtained results.

Author Response

         Please see the attachment.we really appreciated for your detailed recommendations so that the article can be more completed.Also thanks for what you have done for the paper.

Best regards to you.

Reviewer 2 Report

A new correction theory has been put forward by the authors to explain the reducing rate distribution for seamless tube stretch-reducing process. The authors have explained well about the theory and supported their claims with experimental results and verification through simulation.

The following points need to be addressed before the publication of this manuscript.

1.       There are several instances in the paper in which the authors have used inappropriate English sentence structure. It is highly recommended that the authors should revise the complete manuscript for correction of English language mistakes.

2.      There are several typing errors, font errors and punctuation errors in the manuscript. The authors must improve the manuscript for removing all such errors.

3.      In Table 1, the digits of adjacent columns are very close to each other due to 5 significant numbers. Reduce the font size to separate the text in adjacent columns or use vertical grid if allowed by the journal format.

4.      Avoid using yellow color in figure 6 for labeling or dimensioning, as it is not clearly visible.

5.      The resolution of Figure 7 and 8 is not according to the Journal guidelines. Please improve the quality of figure 7 and 8 in which the legend is visible clearly.

6.      Please use the symbol “ד instead of *.

7.      ρ1 should be written as “ρ1”, i.e. 1 should be in the subscript.

8.      Font size of most of the mathematical terms is inconsistent within the manuscript. Make it consistent.

9.      Equation (4) seems misaligned in the text.

10.  It is suggested to use insert “Equation” mode from the MS word or use MathType to type your mathematical terms and equations.

11.  In line #262 please write the units of thermal conductivity after its value.

Author Response

we really appreciated for your detailed recommendations so that the article can be more completed. Also thanks for what you have done for our paper.

Best regards to you.

Round 2

Reviewer 1 Report

Now the paper is improved, with the typo errors and grammer, Editor can accept the paper.

Author Response

Dear Reviewer

    Thank you very much for what you have done for the paper. We are very grateful to you for your careful inspection.
 If any question, please contact us freely.

    Best regards